# Fission Yeast Autophagy Machinery

**DOI:** 10.3390/cells11071086

**Published:** 2022-03-24

**Authors:** Dan-Dan Xu, Li-Lin Du

**Affiliations:** 1National Institute of Biological Sciences, Beijing 102206, China; xudandan@nibs.ac.cn; 2Tsinghua Institute of Multidisciplinary Biomedical Research, Tsinghua University, Beijing 102206, China

**Keywords:** autophagy, autophagy machinery, selective autophagy, fission yeast, *Schizosaccharomyces pombe*

## Abstract

Autophagy is a conserved process that delivers cytoplasmic components to the vacuole/lysosome. It plays important roles in maintaining cellular homeostasis and conferring stress resistance. In the fission yeast *Schizosaccharomyces pombe*, autophagy is important for cell survival under nutrient depletion and ER stress conditions. Experimental analyses of fission yeast autophagy machinery in the last 10 years have unveiled both similarities and differences in autophagosome biogenesis mechanisms between fission yeast and other model eukaryotes for autophagy research, in particular, the budding yeast *Saccharomyces cerevisiae*. More recently, selective autophagy pathways that deliver hydrolytic enzymes, the ER, and mitochondria to the vacuole have been discovered in fission yeast, yielding novel insights into how cargo selectivity can be achieved in autophagy. Here, we review the progress made in understanding the autophagy machinery in fission yeast.

## 1. Introduction

Macroautophagy (hereafter autophagy) is an important cellular degradation system conserved from yeasts to humans. In response to autophagy induction conditions such as starvation, a double-membraned structure called the autophagosome engulfs the cytosol and organelles and delivers them into the vacuole/lysosome for degradation and recycling [1,2,3]. In the budding yeast *Saccharomyces cerevisiae*, the model organism where molecular-level studies of autophagy were first initiated, around 40 autophagy-related (*ATG*) genes have been identified [4,5]. A subset of around 20 *ATG* genes encode core Atg proteins, which play pivotal roles in autophagosome biogenesis and are required for both non-selective bulk autophagy and selective autophagy [6]. These core Atg proteins can be classified into several functional groups, including the Atg1 protein kinase complex, the integral membrane protein Atg9, the phosphatidylinositol 3-kinase (PtdIns3K) complex I, the Atg2-Atg18 complex, and conjugation systems for two ubiquitin-like modifiers (namely, the Atg12 conjugation system and the Atg8 conjugation system). The core Atg proteins are recruited to a punctate structure called the phagophore assembly site or the pre-autophagosomal structure (PAS), where autophagosome biogenesis occurs [7].

Many *ATG* genes have orthologs across eukaryotes, indicating that autophagy is an ancient and conserved process. However, there are also differences in the autophagy machinery between species [8,9]. The fission yeast *Schizosaccharomyces pombe* is a widely used model organism that diverged from *S. cerevisiae* about 600 million years ago [10]. Some aspects of autophagy mechanisms in fission yeast are more similar to those in mammals than in budding yeast. For example, there is a homolog of mammalian Atg101 in *S. pombe* but not in *S. cerevisiae* [11,12,13]; Atg11/FIP200 family proteins are essential for bulk autophagy in *S. pombe* and mammals but not in *S. cerevisiae* [14,15]; and ZZ domain-containing selective autophagy receptors exist in *S. pombe* and mammals but not in *S. cerevisiae* [16,17,18]. Fission yeast autophagy factors were first identified by their homology to previously known budding yeast *ATG* genes [19,20] and later also by unbiased genetic screens that revealed additional autophagy factors that cannot be easily identified by homology search [14,21]. Molecular characterization of these factors in the last decade has led to rich mechanistic insights [13,14,15,17,18,21,22,23,24,25]. In addition, structural biology analyses of *S. pombe* autophagy proteins have also shed new light on the molecular machinery of autophagy [12,26,27]. In this review, we summarize the current understanding of autophagy in fission yeast, with a particular emphasis on the autophagy machinery (Table 1).

## 2. Physiological Roles of Autophagy in Fission Yeast

An evolutionarily conserved role of autophagy is to help cells cope with nutrient starvation. In *S. cerevisiae*, autophagy-defective *atg* mutants lose viability within 5 days upon nitrogen starvation [28,29]. In addition, *atg* mutants exhibit defects in sporulation, a process triggered by starvation in budding yeast [28,30]. Therefore, autophagy is critical for cells to adapt to nutrient-limited conditions.

In the fission yeast, Nakashima et al. reported in 2006 that the vacuolar serine protease Isp6, which is critical for turning over cytoplasmic materials delivered into vacuoles by autophagy, is required for large-scale protein degradation during starvation, and *isp6Δ* fails to initiate mating, a process triggered by nitrogen starvation in fission yeast [31]. In 2007, Kohda et al. reported that fission yeast *atg1*, *atg8,* and *atg13* deletion mutants lose viability during nitrogen starvation and exhibit a mating defect under standard nitrogen-depleted mating conditions [19]. Such a mating defect is shared by the deletion mutants of other fission yeast autophagy genes [14]. Like in budding yeast, *atg* mutants in fission yeast exhibit sporulation deficiency [20]. As both the mating defect and the sporulation defect can be suppressed by providing nitrogen supply [14,19,20], the main role of autophagy in these processes is likely to be supplying a nitrogen source for protein synthesis. In 2020, Zhao et al. reported that autophagy and in particular ER-phagy is important for fission yeast cells to survive against the ER stress inducer dithiothreitol (DTT), but the reason behind this role of autophagy is unclear [25].

## 3. Autophagy Induction Conditions in Fission Yeast

In both budding yeast and fission yeast, autophagy is strongly up-regulated by nitrogen starvation [19,20,31,32]. Carbon starvation can also induce autophagy in budding yeast [32]. However, Kohda et al. showed that carbon starvation does not induce autophagy in fission yeast [19].

In budding yeast, rapamycin is widely used to induce autophagy under nutrient-rich conditions [33]. Both nitrogen starvation and rapamycin treatment result in the inhibition of the TORC1 kinase complex, which negatively regulates autophagy. It has been shown that Atg13, which is a subunit of the Atg1 protein kinase complex, is a key autophagy-related substrate of TORC1 in budding yeast [34,35]. Fission yeast is relatively insensitive to rapamycin treatment, but when rapamycin is applied together with caffeine, which may also inhibit TORC1 activity, autophagy is induced in fission yeast [36,37]. Fission yeast Atg13 is a TORC1-dependent phosphorylation target [38], suggesting that fission yeast and budding yeast may share similar autophagy induction mechanisms.

Apart from nitrogen starvation and rapamycin plus caffeine, the ER stress inducer DTT and the oxidative stress inducer paraquat can also trigger autophagy in fission yeast [25,39], but the induction mechanisms are unclear. Recently, it was reported that sulfur depletion can induce autophagy in fission yeast and that Ecl1 family genes are involved [40].

## 4. Methods to Monitor Autophagy in Fission Yeast

Several methods have been used to monitor the autophagy process in fission yeast. The most classic and commonly used method for assessing autophagy flux is the GFP-Atg8 processing assay. Atg8 is a ubiquitin-like protein playing a central role in autophagosome biogenesis [41]. During autophagy, Atg8 is conjugated to phosphatidylethanolamine (PE) on the autophagic membrane and Atg8 molecules associated with the inner membrane of the autophagosome end up in the vacuole lumen when autophagosome-vacuole fusion occurs [42,43,44]. Because GFP but not Atg8 is resistant to vacuolar proteases, the conversion of the GFP-Atg8 fusion protein to free GFP can be used as a readout for autophagy [45]. Mukaiyama et al. first applied the immunoblotting-based GFP-Atg8 processing assay in fission yeast [20]. Thereafter, the processing of GFP-Atg8, CFP-Atg8, or YFP-Atg8 has been widely used to monitor bulk autophagy in fission yeast [14,15,25].

Based on the same principle, autophagy flux can be assayed using other proteins fused with a fluorescent protein. Tdh1, the major form of glyceraldehyde-3-phosphate dehydrogenase (GAPDH) in fission yeast, is an abundant cytosolic protein [46]. During bulk autophagy, Tdh1 is non-selectively engulfed into the autophagosome and delivered into the vacuole. Therefore, the processing of Tdh1-YFP to free YFP has been used as a readout for bulk autophagy [21,22]. In much the same way, ER-phagy has been monitored using the processing of fluorescent protein-fused integral ER membrane proteins including Ost4 and Erg11, whereas mitophagy has been assayed using the processing of fluorescent protein-fused mitochondrial matrix proteins including Sdh2 and Tuf1 [22,24,25].

A more quantitative method for measuring autophagy flux is the Pho8Δ60 assay [47,48]. *S. cerevisiae* Pho8 is a vacuolar alkaline phosphatase with a N-terminal transmembrane domain. It is synthesized as an inactive precursor and transported from the ER to the vacuole via the secretory pathway. Once entering into the vacuole, the Pho8 protein is processed by vacuolar proteases, resulting in the activation of its phosphatase activity [49]. A truncated form of Pho8, Pho8Δ60, which lacks the N-terminal transmembrane domain, is a soluble cytosolic protein, is only delivered into the vacuole, and becomes activated through bulk autophagy. The autophagy flux can therefore be monitored by examining the Pho8Δ60 activity [47]. Yu et al. adopted this assay for fission yeast by expressing *S. cerevisiae* Pho8Δ60 in an *S. pombe pho8Δ* strain background [21].

The autophagy process can also be assessed by examining the subcellular localization of fluorescent protein-tagged Atg proteins. In budding yeast, most Atg proteins are transiently associated with the PAS during autophagy and form visible puncta [6,7]. Atg8 was the first Atg protein shown to form puncta during autophagy in fission yeast and has been used as a marker for the PAS [19,20]. Sun et al. comprehensively examined the localization of fission yeast Atg proteins and found that during nitrogen starvation-induced autophagy, most of them co-localize at the PAS with Atg8 [14].

Transmission electron microscopy (TEM) is an important technique for visualizing autophagosomes and is particularly useful for quantifying the number and size of autophagosomes [50]. Because autophagosomes are transient structures, visualization of autophagosomes is usually performed in a mutant background where autophagosome-vacuole fusion is blocked and autophagosomes accumulate in the cytoplasm. In fission yeast, Sun et al. reported that deletion of the *fsc1* gene blocks autophagosome-vacuole fusion, and as a result autophagosomes accumulate upon nitrogen starvation [14]. *fsc1Δ* background has been used for TEM-based analysis of autophagosomes [21,22]. Another method that has been used to visualize autophagosomes is the fluorescence loss in photobleaching (FLIP) assay [14]. In this assay, repetitive photobleaching of a small region near one tip of the cell results in the loss of the fluorescence signal of the diffusible cytosolic Tdh1-YFP, whereas the fluorescence signal of the Tdh1-YFP protein enclosed by autophagosomes persists [14]. The FLIP assay and the TEM assay have complementary strengths, with the former being performed on live cells and thus free of fixation artifacts and the latter offering a higher spatial resolution. These two assays have been used together to analyze the size and number of autophagosomes in fission yeast [21,22].

## 5. The Molecular Machinery for Autophagosome Assembly in Fission Yeast

In fission yeast, the core Atg proteins responsible for autophagosome biogenesis can also be classified into six functional groups: the Atg1 complex, two integral membrane proteins Atg9 and Ctl1, the phosphatidylinositol 3-kinase (PtdIns3K) complex I, Atg18 family proteins and Atg2, and Atg12 and Atg8 conjugation systems (Figure 1). Here, we review our understanding of each functional group and discuss the conservation and divergence of autophagy mechanisms between *S. pombe*, *S. cerevisiae*, and mammals.

### 5.1. The Atg1 Protein Kinase Complex

The Atg1 protein kinase complex is critical for autophagy initiation. In *S. cerevisiae*, the Atg1 complex functioning in bulk autophagy is composed of Atg1 serine/threonine protein kinase, the scaffold protein Atg13, and the Atg17-Atg31-Atg29 sub-complex [51,52,53]. Atg11 is a subunit only required for selective autophagy but not for bulk autophagy [54]. Under nutrient-rich conditions, active TORC1 phosphorylates Atg13 to reduce its interaction with Atg1 and Atg17-Atg31-Atg29, thus blocking autophagy [34,35,55,56]. When TORC1 activity is inhibited upon nutrient deprivation or rapamycin treatment, Atg13 is dephosphorylated to allow its interactions with Atg1 and Atg17, resulting in the assembly of a supramolecular complex, thereby triggering Atg1 auto-phosphorylation and auto-activation [56]. In selective autophagy, Atg11 interacts with Atg1 and autophagy receptors such as Atg19 and Atg32 and tethers the Atg1 kinase complex to multimeric cargo-receptor complexes, thereby promoting Atg1 auto-phosphorylation and auto-activation [57,58]. The mammalian counterpart to the budding yeast Atg1 complex is the ULK1 complex, which consists of the ULK1 kinase, FIP200, ATG13, and ATG101 [11]. Unlike the situation in *S. cerevisiae*, the ULK1 complex is constitutively formed irrespective of whether mammalian TORC1 (mTORC1) is active. But mTORC1 can also modulate ULK1 kinase activity through the phosphorylation of ULK1 and ATG13 [59,60,61].

Like the *S. cerevisiae* Atg1 complex, the *S. pombe* Atg1 complex has Atg1, Atg13, Atg17, and Atg11 subunits. Unlike the *S. cerevisiae* Atg1 complex but similar to the mammalian ULK1 complex, the *S. pombe* Atg1 complex is devoid of Atg29 and Atg31 subunits but contains a homolog of mammalian Atg101 [12,13,14,15,62]. Moreover, *S. pombe* Atg101 has no detectable sequence homology to *S. cerevisiae* Atg29 and Atg31, and *S. pombe* Atg101 cannot functionally substitute for *S. cerevisiae* Atg29 and Atg31, suggesting that Atg101 is not the counterpart of Atg29-Atg31 [12,13]. Analyzing the inter-subunit interactions reveals that, like the situation in *S. cerevisiae*, in *S. pombe*, Atg13 directly interacts with Atg1 and Atg17 [13,62]. Atg1 and Atg17 may also directly interact [62]. Atg101 interacts with Atg13 but not with other subunits [13,62], and Atg101 is responsible for stabilizing Atg13 [12]. Atg11 not only strongly interacts with Atg1 but also weakly interacts with Atg13 [15,62].

Atg11 in *S. cerevisiae* is dispensable for bulk autophagy [63]. However, Atg11 in *S. pombe* is essential for starvation-induced bulk autophagy [14]. FIP200, the ortholog of Atg11 in mammals, is also required for bulk autophagy [60,64]. Importantly, unlike the situation in *S. cerevisiae* where Atg1 kinase activity is regulated by Atg13 and Atg17, in *S. pombe*, Atg11 but not Atg13 and Atg17 is required for the normal Atg1 kinase activity [15]. *S. pombe* Atg11 can homodimerize and thereby bring two Atg1 molecules together so that Atg1 can auto-phosphorylate and become activated [15]. In *S. cerevisiae* and mammalian cells, the activated Atg1/ULK1 can phosphorylate downstream Atg factors, such as Atg4 and Atg9 in *S. cerevisiae* [65,66] and ATG4B, ATG9, ATG14L, and Beclin 1 in mammalian cells [67,68,69,70]. In *S. pombe*, the phosphorylation substrates of the Atg1 kinase other than itself are not yet known.

### 5.2. Atg9 and Ctl1

In *S. cerevisiae*, Atg9 is the sole transmembrane protein among core Atg proteins. Fluorescence microscopy analysis of the localization of *S. cerevisiae* Atg9 shows that Atg9 is not exclusively present at the PAS but also exists in cytoplasmic punctate structures termed non-PAS sites [71,72]. During autophagy, Atg9 shuttles between the PAS and non-PAS sites [71,73]. The anterograde transport of Atg9 from non-PAS sites to the PAS is facilitated by Atg23 and Atg27, which bind Atg9 to form a cycling complex for efficient Atg9 traffic, whereas the retrieval of Atg9 from the PAS to non-PAS sites requires the Atg1-Atg13 complex, Atg2, Atg18, and the PI3K complex I [72,74,75]. In mammals, ATG9A is an ortholog of budding yeast Atg9 [76]. Under nutrient-rich conditions, ATG9A is localized to the trans-Golgi network (TGN) and endosomes, and starvation causes ATG9A to relocalize to the PAS [77].

*S. pombe* Atg9 was first identified based on sequence homology to *S. cerevisiae* Atg9 [20]. Deletion of the *atg9* gene blocks starvation-induced autophagy in *S. pombe* [14,20]. Analysis of YFP-tagged Atg9 shows that, like the situation in *S. cerevisiae*, during autophagy, Atg9 localizes to both the PAS and non-PAS cytoplasmic punctate structures [14]. No obvious sequence homologs of *S. cerevisiae* Atg23 and Atg27 exist in *S. pombe*. However, Sun et al. identified a novel fission yeast autophagy factor Ctl1, which performs a functional role similar to that of *S. cerevisiae* Atg23 and Atg27 [14]. Ctl1 is a multi-transmembrane protein and co-localizes with Atg9 on the PAS and non-PAS compartments. In addition, Ctl1 physically interacts with Atg9, and the loss of one of them affects the localization of the other. No counterparts of Atg23-Atg27 or Ctl1 have been found in mammals. The deletion of *atg1* or *atg2* restricts the localization of both Atg9 and Ctl1 to the PAS, suggesting that their recycling from the PAS to non-PAS compartments relies on Atg1 and Atg2. It remains unclear how Atg1 and Atg2 promote Atg9 retrograde trafficking.

Recently, the structural and biochemical analyses of Atg9 shed new light on its function. The cryo-EM structures of plant, human, and fission yeast Atg9/ATG9A reveal a homotrimeric architecture containing two channels with the potential of transporting lipid [27,78,79,80]. Biochemical analyses showed that Atg9 can transport lipids between the two leaflets of the membrane, and this lipid scramblase activity may help drive the expansion of the phagophore [27,80].

### 5.3. PtdIns3K Complexes

Apart from the Atg1 complex and Atg9, autophagy initiation also requires an autophagy-specific PtdIns3K complex [7,81]. In budding yeast and humans, there exist two PtdIns3K complexes: the PtdIns3K complex I and the PtdIns3K complex II. The former is composed of five subunits: Vps34/VPS34, Vps15/VPS15, Atg6 (also known as Vps30)/Beclin 1, Atg38/NRBF2, and Atg14/ATG14L, and the latter is composed of four subunits: Vps34/VPS34, Vps15/VPS15, Atg6/Beclin 1, and Vps38/UVRAG [82,83,84,85,86]. Both PtdIns3K complexes can phosphorylate the 3-OH position of phosphatidylinositol (PtdIns) to generate phosphatidylinositol-3-phosphate (PtdIns3P) on the target membrane through the lipid kinase activity of Vps34/VPS34, but only the PtdIns3K complex I is essential for the initiation of autophagy, whereas the PtdIns3K complex II plays a role in endocytic trafficking. Their distinct functional roles can be attributed to the difference in their localization: the PtdIns3K complex I is localized to the PAS in budding yeast, where the PtdIns3K complex II is localized to the endosome.

In fission yeast, there are also two PtdIns3K complexes. Identification and characterization of *S. pombe* Vps34 were published in 1995 [87,88]. BLAST search of *S. pombe* homologs of *S. cerevisiae* Atg proteins identified fission yeast Vps15 and Atg6 [89]. In 2013, fission yeast Atg14 and Vps38 were reported based on the results of a genetic screen [14]. The fifth subunit of the PtdIns3K complex I, Atg38, was reported in 2020 [21]. The compositions of PtdIns3K complexes in fission yeast have now been completely identified and are conserved to budding yeast and mammalian complexes. To understand the organizations of the two *S. pombe* PtdIns3K complexes, Yu et al. applied a newly established imaging-based assay for detecting protein–protein interactions, called the Pil1 co-tethering assay, to analyze the interaction relationships between the components of the PtdIns3K complexes [62]. Like the situations in budding yeast and mammals, Vps15 bridges the association between the Atg6-Atg14 subcomplex and Vps34 in the PtdIns3K complex I and the association between the Atg6-Vps38 subcomplex and Vps34 in the PtdIns3K complex II. Intriguingly, how Atg38 (NRBF2 in mammals) is incorporated into the PtdIns3K complex I seems different among species, with fission yeast Atg38 binding to Vps34, budding yeast Atg38 interacting with Atg14, and mammalian NRBF2 interacting with ATG14 and Beclin 1 (homolog of yeast Atg6).

*S. pombe* Atg38 possesses an Atg8-family-interacting motif (AIM, also called LC3-interacting region, LIR) [21]. Through the Atg8–Atg38 interaction, a positive feedback loop is established between Atg8 and the PtdIns3K complex I to enhance the PAS accumulation of the PtdIns3K complex I and autophagy factors acting downstream of this complex [21]. Although the AIM is only present in fission yeast Atg38 but not in budding yeast Atg38 and mammalian NRBF2, these non-fission-yeast species may establish the connections between Atg8 and the PtdIns3K complex I through Atg8 binding to other subunits of the PtdIns3K complex I. For example, mammalian ATG14L has been reported to harbor an LIR within its BATS domain, which mediates its interaction with Atg8-family proteins GABARAP and GABARAPL1 [90], suggesting that AIM-mediated connections between Atg8 and the PtdIns3K complex I are a common autophagy mechanism.

### 5.4. Atg18 Family Proteins and Atg2

Subsequent to PtdIns3P generation by the PtdIns3K complex I on the PAS, the Atg18 family proteins, which are PtdIns3P effectors, are recruited to the PAS through binding PtdIns3P and function as platforms for recruiting downstream Atg proteins. In *S. cerevisiae*, there are three Atg18 family proteins: Atg18, Atg21, and Hsv2 [91]. Only Atg18 is essential for both non-selective bulk autophagy and selective autophagy [92,93]. Atg21 functions mainly in the Cvt pathway, a selective autophagy pathway [94]. Atg18 interacts with Atg2 to promote phagophore expansion [7,95,96], whereas Atg21 interacts with Atg16 and Atg8 to promote Atg8 lipidation [97]. Mammals have four Atg18 family proteins (WIPI1–4) [98]. All mammalian WIPI proteins localize to autophagosome membranes upon nutrient starvation [99]. WIPI2 promotes PE conjugation of LC3 by recruiting the ATG12-ATG5·ATG16L complex via a direct interaction with ATG16L [100]. WIPI1 interacts with WIPI2, and this interaction is proposed to facilitate ATG16L complex recruitment [99]. WIPI3 and WIPI4 are scaffolds to link upstream regulatory pathways to PtdIns3P production by interacting with the TSC complex and the ULK1 complex, respectively, in response to glucose starvation [99]. Similar to Atg18 in *S. cerevisiae*, WIPI4 also interacts with ATG2 to promote phagophore expansion [101,102]. All Atg18/WIPI proteins are targeted to autophagic membranes through binding PtdIns3P, and their PtdIns3P binding activities are required for autophagy [95,103]. Structural analyses revealed that Atg18/WIPI proteins fold into a seven-bladed β-propeller with two PtdIns3P binding sites locating in the conserved Phe-Arg-Arg-Gly (FRRG) motif at blade 5 and blade 6, and the binding sites for Atg2 and Atg16 are located at the opposite surface from the FRRG motif [104,105,106].

Fission yeast also has three Atg18 family proteins, Atg18a, Atg18b, and Atg18c [14,20]. Unlike the situation in *S. cerevisiae*, these three Atg18 family proteins are all required for bulk autophagy [14]. Upon nitrogen starvation, Atg18a, Atg18b, and Atg18c form puncta co-localizing with Atg8 on the PAS [14]. In addition, Atg18a is also observed on the vacuole membrane and endosomes under nutrient-rich conditions. The proper localization of these three Atg18 family proteins relies on their conserved FRRG motifs. Mutating the FRRG motif to FTTG makes Atg18a diffusely distributed and blocks autophagy [14]. Analyzing the localization of CFP-Atg8 in *atg* mutants during starvation, Sun et al. found that *atg18aΔ* abolished Atg8 puncta formation, whereas *atg18bΔ*, *atg18cΔ*, and *atg2Δ* elevated the levels of Atg8 puncta [14], suggesting that Atg18a may play a role different from that of the other two Atg18 family proteins and that Atg18b, Atg18c, and Atg2 may function in a similar way. Further dissecting the molecular role of Atg18a showed that Atg18a serves as a binding platform for the recruitment of the Atg12-Atg5·Atg16 complex to the PAS [14].

Recent structural and biochemical analyses reveal the molecular roles of Atg2. Electron microscopy analyses of human ATG2A and ATG2B as well as budding yeast Atg2 uncover a rod-like shape of these proteins [101,102]. Atg2 itself has membrane-binding ability by using two independent binding regions at both ends of the rod structure [107,108]. The N-terminal region of Atg2 is responsible for the ER localization, while a C-terminal amphipathic helix is required for its localization to the phagophore [108]. Both membrane-binding regions are critical for autophagy. In vitro experiments revealed that Atg2 can tether small liposomes [102,108]. These observations suggested that Atg2 has membrane-tethering activity to tether the ER membrane to the phagophore. In line with this, Atg2-Atg18 has been shown to localize at the edge of the expanding phagophore and co-localize with ER exit sites (ERES) in yeast [109,110]. In addition to the membrane-tethering activity, Atg2 can transfer lipids from ER to the phagophore. The initial hint was provided by the crystal structure of the N-terminal region of fission yeast Atg2, which shows a tubular fold with a hydrophobic cavity that can accommodate tens of glycerophospholipid molecules [26], and in vitro analysis showed that Atg2 can transfer phospholipids with little specificity for the head group [26,111]. Osawa et al. showed that the membrane-tethering activity and lipid transfer activity of human ATG2B are accelerated by WIPI4 and PtdIns3P [112]. Fission yeast Atg18b, Atg18c, and Atg2 localize at the tips of phagophores [14]. Further studies will be needed to determine whether Atg18b and Atg18c facilitate the membrane-tethering activity and the lipid transfer activity of Atg2, and why two different Atg18 family proteins are needed to act with Atg2 in fission yeast.

### 5.5. Two Ubiquitin-like Conjugation Systems

Two ubiquitin-like conjugation systems are the most downstream autophagy factors. Among them, Atg12/ATG12 and Atg8-family proteins (Atg8 in yeast and microtubule-associated protein light chain 3 (LC3) isoforms and gamma-aminobutyric acid receptor-associated proteins (GABARAPs) in mammals) are ubiquitin-like proteins and are conjugated to Atg5/ATG5 and phosphatidylethanolamine (PE), respectively [42,113]. Atg12/ATG12 is C-terminally conjugated to Atg5/ATG5 through the successive enzymatic reactions involving the E1-like enzyme Atg7/ATG7 and the E2-like enzyme Atg10/ATG10 [113,114]. The Atg12-Atg5/ATG12-ATG5 conjugate interacts with Atg16/ATG16 to form the Atg12-Atg5·Atg16 complex [115,116].

In the Atg8-PE conjugation system, Atg8-family proteins are firstly cleaved by the Atg4/ATG4 cysteine protease to expose their C-terminal glycine residue [42]. Then, Atg7/ATG7, Atg3/ATG3, and the Atg12-Atg5/ATG12-ATG5 conjugate act as E1, E2, and E3 enzymes, respectively, to promote the conjugation of Atg8 to PE [42,117]. Although Atg16/ATG16 forms complexes with Atg12-Atg5/ATG12-ATG5, it is dispensable for the E3-like enzyme activity of Atg12-Atg5/ATG12-ATG5 conjugate but specifies the sites of Atg8-PE formation on the autophagy-related membranes [118].

In fission yeast, Atg3, Atg4, Atg5, Atg7, Atg8, and Atg12 were firstly found through homology search [19,20], while Atg10 and Atg16 were identified in a genetic screen [14]. All these proteins are required for autophagy. Deleting *atg3*, *atg4*, *atg5*, *atg7*, *atg10*, *atg12*, or *atg16* blocks the puncta formation of Atg8 and inhibits the processing of CFP-Atg8 [14]. Atg8 is commonly regarded as the most downstream Atg protein. However, as mentioned earlier, there is a positive feedback loop between Atg8 and the PtdIns3K complex I via the Atg8-Atg38 interaction [21], suggesting that Atg protein recruitment to the PAS is not simply in a hierarchical manner.

## 6. Selective Autophagy Receptors in *S. pombe*

Autophagy not only can nonspecifically engulf the cytoplasm into autophagosomes for degradation but also can specifically target certain cytoplasmic components for delivery into the lysosome/vacuole. The selectivity is achieved by autophagy receptors (also called autophagy adaptors). Here, we review the currently known selective autophagy receptors in fission yeast (Figure 2).

The first identified selective autophagy pathway is the budding yeast cytoplasm-to-vacuole targeting (Cvt) pathway, in which the aminopeptidase Ape1 is constitutively transported into the vacuole utilizing the autophagy machinery [119]. Ape1 is first synthesized in the cytoplasm and selectively recognized by the receptor protein Atg19, which in turn interacts with Atg8 and the scaffold protein Atg11, thus connecting the Ape1 cargo to the core autophagy machinery [54,120]. Although mammalian autophagy receptor NBR1 is postulated to be distantly related to Atg19 [16], in mammals, NBR1 has not been implicated in hydrolase transport but instead participates in the selective autophagy of aggregated proteins, midbodies, and peroxisomes [121,122,123]. In fission yeast, through analyzing Nbr1, a sequence homolog of mammalian NBR1, Liu et al. found that Nbr1 mediates vacuolar targeting of two aminopeptidases, Lap2 and Ape2, and named this selective autophagy pathway the Nbr1-mediated vacuolar targeting (NVT) pathway [17]. Thereafter, Wang et al. identified two additional NVT cargos, the mannosidase Ams1 and the aminopeptidase Ape4 [18]. Unlike the Cvt pathway in budding yeast, the NVT pathway is dependent on the endosomal sorting complexes required for transport (ESCRTs) but not the macroautophagy machinery, suggesting a novel mode of action of autophagy receptors [17]. Cargo selectivity is achieved by the specific interactions between hydrolase cargos and three ZZ domains in Nbr1, with the ZZ1 domain binding Ams1 and Ape4 and the ZZ2 and ZZ3 domains together recognizing Lap2 and Ape2 [17,18]. Cryo-EM structures of the Nbr1-Ams1 complex and the Nbr1-Ape4 complex have provided atomic level insights into how a single ZZ domain can bispecifically recognize two distinct protein cargos [18].

Apart from protein cargos, selective autophagy can also target organelles, including the endoplasmic reticulum (ER), mitochondria, peroxisomes, lysosomes, lipid droplets, and nucleus [124]. Autophagy receptors involved in organelle autophagy are usually Atg8/LC3-binding proteins and harbor the AIM/LIR sequences [125]. In fission yeast, two types of organelles, the ER and the mitochondrion, are degraded by autophagy during starvation in a manner requiring the general autophagy machinery and the Atg20- and Atg24-family proteins [22,126]. The Atg20- and Atg24-family proteins may contribute to organelle autophagy by promoting the formation of autophagosomes of sufficient sizes [22]. It was only recently that the specific receptors dictating the selectivity of autophagic degradation of these two types of organelles in fission yeast were identified [24,25].

Based on the assumption that selective autophagy receptors interact with Atg8, Liu et al. performed affinity-purification coupled with mass spectrometry analysis using Atg8 as bait. Besides identifying a vacuole membrane protein Hfl1 as an Atg8-binding protein involved in the lipidation-independent vacuolar functions of Atg8, they also identified an uncharacterized protein, Mug185, as an Atg8 interactor [23]. Hereafter, Zhao et al. analyzed the functional roles of Mug185 and showed that it is an ER-phagy receptor, thus renaming it Epr1 [25]. Epr1 can simultaneously interact with Atg8 and VAP-family integral ER membrane proteins through its AIM and FFAT motifs, respectively, and thereby act as a bridging molecule to mediate the association between the ER and the phagophore. In addition, the ER-plasma membrane contact supported by VAPs and Ire1-dependent Epr1 upregulation are also important for ER stress-induced ER-phagy. Loss of either Epr1 or VAPs reduces cell survival against ER stress, whereas bypassing their ER-phagy functions by artificial tethers can rescue the survival, indicating that Epr1- and VAP-mediated ER-phagy is of physiological importance. Prior to the publication of the Epr1 study, many ER-phagy-specific autophagy receptors have been identified, including Atg39 and Atg40 in budding yeast [127], and RETREG1/FAM134B, SEC62, RTN3L, CCPG1, TEX264, and ATL3 in mammals [128,129,130,131,132,133,134]. All of them are integral ER membrane proteins. In contrast to these receptors, Epr1 is a soluble protein and localizes to the ER through interacting with VAPs [25]. Parallel to this work, a soluble autophagy receptor CALCOCO1 was shown to act with VAPs to mediate ER-phagy in mammals, suggesting a similarity in ER-phagy mechanisms between fission yeast and mammals [135]. In Epr1, the C-terminal region containing AIM and FFAT motifs is not conserved outside of the fission yeast lineage, suggesting that autophagy receptor functions, which are conferred by short linear motifs, can be acquired or lost relatively frequently during evolution.

Mitophagy is thought to play an important role in quality and quantity control of mitochondria [136,137]. In mammals, selective removal of dysfunctional or excessive mitochondria can be achieved by receptor-mediated pathways, where multiple mitochondrial outer membrane (MOM) proteins, including NIX, BNIP3, FUNDC1, BCL2-L-13, and FKBP8, have been identified as receptors directly binding LC3/GABARAP via LIR motifs to promote mitophagy [138], as well as ubiquitination-dependent pathway, in which MOM proteins are ubiquitylated by Parkin and bind LIR-containing ubiquitin-binding adaptors, such as OPTN and NDP52 [139]. In budding yeast, Atg32 is the sole identified mitophagy receptor, which localizes to the MOM [140,141]. In fission yeast, Atg43 is recently identified as a mitophagy receptor through screening the *S. pombe* deletion library [24]. Atg43 is a transmembrane protein anchored to the MOM. It harbors an AIM motif in its cytosolic region. Artificially tethering Atg8 to mitochondria can bypass the mitophagy function of Atg43, suggesting that the primary role of Atg43 in mitophagy is to connect Atg8 to the MOM. In addition, mitochondrial import factors, including the MIM complex and Tom70, are also essential for mitophagy, with the MIM complex facilitating the integration of Atg43 into the MOM, and Tom70 promoting mitophagy via an unknown mechanism. Disrupting the mitophagy function of Atg43 reduces cell survival during nitrogen starvation and affects mitochondrial oxidative status, suggesting Atg43-mediated mitophagy is of physiological significance. Apart from acting as a mitophagy receptor, Atg43 also has a mitophagy-independent function in promoting vegetative growth. The AIM-containing region in Atg43 is not conserved outside of the *Schizosaccharomyces* species, suggesting that like Epr1, Atg43 may have acquired autophagy functions after the divergence of fission yeasts from other fungi.

## 7. Unanswered Questions

Despite intensive research efforts over more than two decades, our understanding of the molecular mechanisms of autophagy remains incomplete. Compared to budding yeast and mammals, fission yeast is an under-utilized model for autophagy research, and therefore there are many open questions awaiting to be answered. For example, the only known substrate of the Atg1 kinase in fission yeast is Atg1 itself [15]. Identifying additional substrates of Atg1 will provide insights into how autophagy machinery is regulated, and in particular how Atg9 retrograde transport is controlled by Atg1. The proper targeting of Atg9 to the PAS is facilitated by its binding partner Ctl1 [14]. How Ctl1 promotes the PAS targeting of Atg9 and whether there are additional factors regulating Atg9 remain unclear. Atg14 determines the PAS localization of the PtdIns3K complex I [21], but the mechanistic basis is unknown. The conserved “cysteine repeats” in the N-terminal region of Atg14 may be involved [14] and are worth investigating. Most eukaryotic species have more than one member of the Atg18 family proteins. All three Atg18 family proteins in fission yeast, Atg18a, Atg18b, and Atg18c, are essential for autophagy [14], and thus fission yeast is a useful model for understanding the functional specialization of Atg18 family proteins. Our knowledge on selective autophagy in fission yeast is still very limited. Like other organisms, ER-phagy and mitophagy in fission yeast rely on autophagy receptors [24,25]. It will be interesting to identify additional factors involved in ER-phagy and mitophagy. Whether other selective autophagy processes, including pexophagy, lipophay, and ribophagy, exist in fission yeast is another interesting question waiting to be addressed.

## 8. Conclusions

Autophagy research in *S. pombe* has made great progress in the past decade, especially in the identification and characterization of both general autophagy factors and selective autophagy factors. Comparing the molecular mechanisms of autophagy between the fission yeast, budding yeast, and mammals revealed the basic machinery of autophagy is conserved among these species. However, there also exist many differences, and some mechanisms in fission yeast are proved to be more similar to mammals than to budding yeast. Further studies on autophagy using fission yeast as a model organism are expected to offer more insights into the universality and diversity of autophagy mechanisms.

## Figures and Tables

**Figure 1 cells-11-01086-f001:**
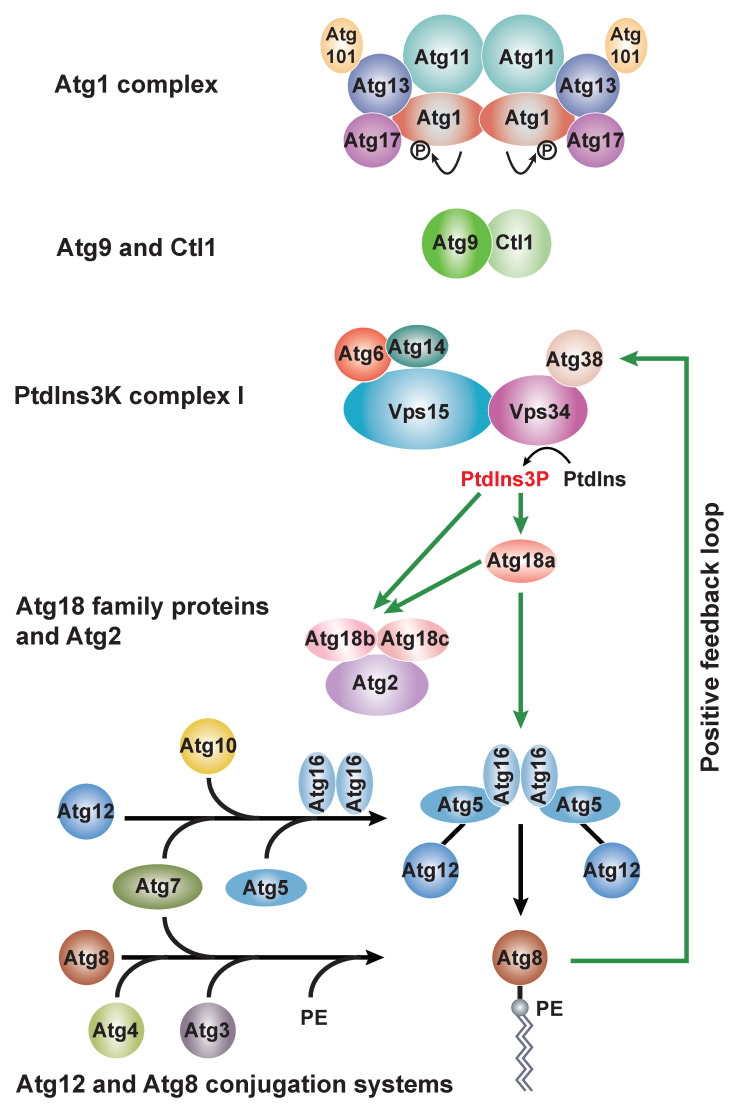
Six functional groups involved in autophagosome biogenesis in fission yeast. The Atg1 complex, Atg9 and Ctl1, and the PtdIns3K complex I are important for autophagy initiation and act upstream of other autophagy factors. The lipid kinase Vps34 catalyzes the conversion of phosphatidylinositol (PtdIns) to phosphatidylinositol-3-phosphate (PtdIns3P) at the PAS. PtdIns3P effectors Atg18a, Atg18b, and Atg18c are recruited to the PAS. Atg18a promotes the PAS targeting of Atg18b, Atg18c, and Atg2 (our unpublished results). Atg18a also recruits the Atg12-Atg5·Atg16 complex to the PAS to promote the conjugation of Atg8 to phosphatidylethanolamine (PE). Atg8 interacts with Atg38 to establish a positive feedback loop between Atg8 and the PtdIns3K complex I. Green arrows denote PAS recruitment. Black arrows denote promotion of Atg12 or Atg8 conjugation.

**Figure 2 cells-11-01086-f002:**
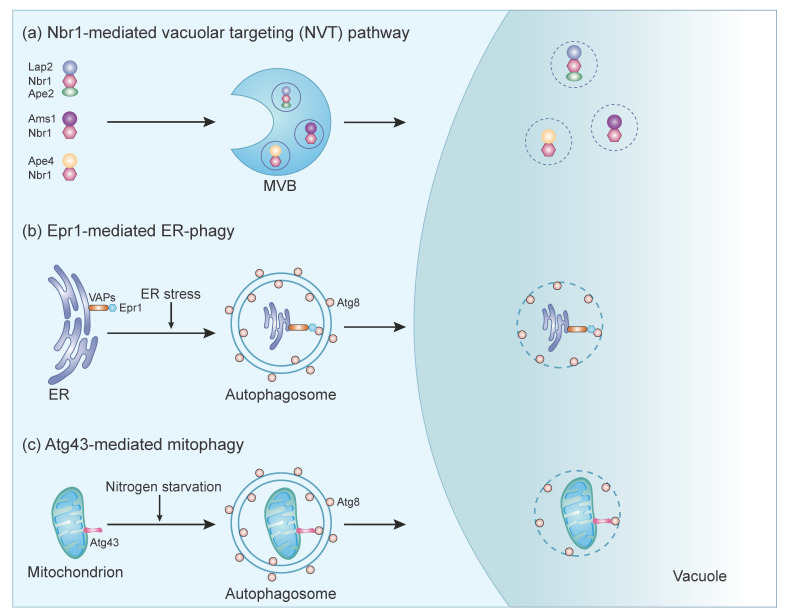
Three types of selective autophagy in fission yeast. (**a**) NVT cargos form three separate complexes with Nbr1 and are delivered into the vacuole through the multi-vesicular body (MVB). (**b**) The soluble ER-phagy receptor Epr1 localizes on the ER through interacting with integral ER membrane proteins VAPs. Under ER stress conditions, the Epr1–Atg8 interaction promotes the engulfment of ER fragments into autophagosomes. (**c**) Atg43 is a mitochondrial outer membrane protein. During nitrogen starvation, the Atg43-Atg8 interaction promotes the engulfment of mitochondria into autophagosomes.

**Table 1 cells-11-01086-t001:** Autophagy factors in *S. pombe*.

*S. pombe* Autophagy Factor	Function in *S. pombe*	*S. cerevisiae* Homolog Involved in Autophagy	Human Homolog Involved in Autophagy
The Atg1 complex			
Atg1	Serine/threonine kinase [15]	Atg1	ULK1
Atg11	Dimerizing Atg1 and activating Atg1 activity [15]	Atg11	FIP200
Atg17	Component of the Atg1 complex [13]	Atg17	−
Atg13	Component of the Atg1 complex [13]	Atg13	ATG13
Atg101	Stabilizing Atg13 [12,13]	−	ATG101
Atg9 and Ctl1			
Atg9	Lipid scramblase [27]	Atg9	ATG9A/B
Ctl1	Promoting proper localization of Atg9 [14]	−	−
The PtdIns3K complex I			
Vps34	PtdIns 3-kinase	Vps34	VPS34
Vps15	Protein kinase required for Vps34 activity	Vps15	VPS15
Atg14	Autophagy specific subunit of PtdIns3K complex I [14]	Atg14	ATG14
Atg6	Forming a subcomplex with Atg14 [14]	Atg6	Beclin 1
Atg38	Interacting with Atg8 to establish a positive feedback loop [21]	Atg38	NRBF2
Atg18 family proteins and Atg2			
Atg2	Membrane tethering and lipid transfer [26]	Atg2	ATG2A/B
Atg18a	Targeting the Atg12-Atg5·Atg16 complex to the PAS [14]	Atg21	WIPI2
Atg18b	Acting with Atg18c and Atg2 [14]	Atg18	WIPI4
Atg18c	Acting with Atg18b and Atg2 [14]		
Atg12 and Atg8 conjugation systems			
Atg12	Ubiquitin-like protein conjugated to Atg5 [14,20]	Atg12	ATG12
Atg7	E1 enzyme for Atg12 and Atg8 [14,20]	Atg7	ATG7
Atg10	E2 enzyme for Atg12 [14]	Atg10	ATG10
Atg5	Acting with Atg12 as the E3 enzyme for Atg8 [14,20]	Atg5	ATG5
Atg16	Promoting the PAS localization of Atg12-Atg5 [14]	Atg16	ATG16L1/2
Atg8	Ubiquitin-like protein conjugated to PE [14,20]	Atg8	LC3 and GABARAP
Atg4	Cysteine protease for Atg8 C-terminal processing and delipidation [14,20]	Atg4	ATG4A-D
Atg3	E2 enzyme for Atg8 [14,20]	Atg3	ATG3
Proteins involved in selective autophagy			
Nbr1	Targeting cytosolic hydrolases to the vacuole [17,18]	Atg19	NBR1
Atg20	Atg20 family protein promoting ER and mitochondrial autophagy [22]	Atg20	−
Atg24 and Atg24b	Atg24 family proteins promoting ER and mitochondrial autophagy [22]	Atg24	−
Epr1	ER-phagy receptor [25]	−	−
Atg43	Mitophagy receptor [24]	−	−

## Data Availability

This article has no additional data.

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
