# Peer review of "Fission Yeast Autophagy Machinery"

_cells, 2022, doi:10.3390/cells11071086_

Round 1
Reviewer 1 Report
In this review article, Xu and Du introduced the fission yeast factors involved in autophagy and compared the machinery with budding yeast and mammals. The manuscript is well written. However, it becomes even better if the authors can clarify current important questions in this field and how the fission yeast study will contribute to understanding the mechanism of autophagy.
Major points:
- It is better to discuss more the mechanism depicted in Figure 1 and discuss important questions in this field.
- Figure 1. It isn’t easy to understand the link between different complexes. It is better to indicate what each arrow means (e.g., the arrow from “Atg1 complex” to “Atg9 and Ctl1”).
- Lines 230-232. “The deletion of atg1 or atg2…on Atg1 and Atg2.” Can the authors explain how Atg1 and Atg2 effect Atg9 localization?
- Lines 248-250. “but only the PtdIns3K complex I is … complex II has a role in endocytic trafficking.” Can the authors explain or discuss what makes the functional difference between complex I and II.
- Line 268. “a positive feedback loop” It is hard to understand the mechanism of this positive feedback loop. Does the interaction between Atg38 and Atg8 affect their intracellular localization or stimulate Vsp34 activity?
Minor points:
- Lines 124-126. “Yu et al. expressed S. cerevisiae … in fission yeast [21].” It is unclear how budding yeast Pho8∆60 activity in fission yeast pho8∆ cells can be a readout of autophagy.
- Line 152-154. “the Atg1 protein kinase … Atg12 and Atg8” It is better to use the consistent nomenclatures of these functional groups here, Figure 1, and Table 1.
- Figure 1. Some of the colors are too dark to see the protein names.
- Figure 1. It is better to include Atg4 in this figure.
- Figure 1 legend. Please explain the following abbreviations, PI, PI3P, and PE. Are PI3P and PtdIns3P the same thing? If so, it is better to express it consistently throughout the manuscript.
- Line 194. “Atg1 and Atg17 may also directly interact [62]” But, this is inconsistent with the illustration shown in Figure 1. The authors can modify the image to make it consistent with this and other statements.
- Line 226. Is there a Ctl1 or Atg23-27 counterpart present in mammals? Or, do other factors translocate Atg8 to PAS in mammals?
- Lines 235-238. “These channels mediate lipid transportation…membrane.” This sentence is unclear to non-specialists.
- Lines 448-451. “The region…from other fungi.” I wonder whether Epr1- and Atg43-binding domains of Atg8 have been co-evolved with Epr1 and Atg43?
- Lines 401-402. Can the authors explain details of the Atg20- and Atg24-family proteins?
- Lines 461-461. I cannot find any supplementary materials in the text.
Reviewer 2 Report
Dear Editor, dear authors,
This review summarizes very well the similarities and differences in autophagy in S.pombe, S.cereviaiae and H. sapiens.
Nonetheless some improvements are always possible:
Table1: This table would benefit from ciations. At least citations for each S. pombe autophagic gene should be added.
At some (many) instances technical/biological terms not precisely defined (e.g. C3 symmerty (line 235), Pil1 cothetering assay (line 259), FRRG motif (line 297) and so on).
At the end the review ends very abrupt. It is not clear to me if some auotphagic processes as pexophagy, lipophay, ribophagy.... do not exist in S,pombe, are not identified yet or were not described. This should be clarified and eventually (if possible) described.
